# Benchmarking Acidic and Basic Catalysis for a Robust Production of Biofuel from Waste Cooking Oil

**Claudia Carlucci [1],\*, Michael Andresini [1], Leonardo Degennaro [1] and Renzo Luisi [1,2]** 

[1]  Flow Chemistry and Microreactor Technology FLAME-Lab, Department of Pharmacy-Drug Sciences, University of Bari "A. Moro" Via E. Orabona 4, Bari 70125, Italy; m.andresini1@studenti.uniba.it (M.A.); leonardo.degennaro@uniba.it (L.D.); renzo.luisi@uniba.it (R.L.)

[2]  Institute for the Chemistry of OrganoMetallic Compounds (ICCOM), National Research Council (CNR), Via Orabona 4, Bari 70125, Italy

\*  Correspondence: claudia.carlucci@uniba.it; Tel.: +39-080-544-2251

**Abstract:** The production of biodiesel at the industrial level is mainly based on the use of basic catalysts. Otherwise, also acidic catalysis allowed high conversion and yields, as this method is not affected by the percentage of free fatty acids present in the starting sample. This work has been useful in assessing the possible catalytic pathways in the production of fatty acid methyl esters (FAMEs), starting from different cooking waste oil mixtures, exploring particularly acidic catalysis. It was possible to state that the optimal experimental conditions required concentrated sulfuric acid 20% *w/w* as a catalyst, a reaction time of twelve hours, a temperature of 85 °C and a molar ratio MeOH/oil of 6:1. The role of silica in the purification method was also explored. By evaluating the parameters, type of catalyst, temperature, reaction time and MeOH/oil molar ratios, it has been possible to develop a robust method for the production of biodiesel from real waste mixtures with conversions up to 99%.

**Keywords:** transesterification; catalysis; acidity; waste cooking oil; biodiesel

## 1. Introduction

The World Energy Forum has estimated that fossil fuels will run out in less than a century if they are not replaced by an alternative energy source. Emissions from combustion of fossil fuels, containing $CO_2$, CO, nitrogen and sulfur oxides, organic and particulate compounds, as well as being extremely harmful, are the main cause of the greenhouse effect and acid rainfall. Therefore, alternative energy resources such as solar energy, geothermal energy and biomass have captured the attention of the scientific community, including biodiesel, which is a widely used energy source. The valorization of waste materials is a viable alternative to traditional disposal systems, including in the field of renewable energy and bio-fuels [1]. The discovery of alternative fuels that can replace conventional resources has therefore become a goal of scientific research [2]. Biodiesel is a non-oil variant, consisting of a mixture of fatty acid methyl esters (FAMEs), obtained through the transesterification reaction of triglycerides catalyzed by acids, bases, enzymes [3–5] or by using supercritical fluids [6–9], starting from vegetable oils, fats, algae and all kinds of suitable raw materials and waste products [10], including recycled frying oil. Therefore, any source of fatty acids can be used to produce the biodiesel [11,12]. The advantages related to its use as fuel are numerous, starting from the low environmental impact, as it represents a renewable and biodegradable source of energy. The current trend is not to use edible raw materials preferring waste ones, which still must comply with the quality standards as essential prerequisites for the commercial use of any fuel [13–16]. The disposal of waste vegetable oil represents an important issue, especially due to the widespread incorrect dumping in urban wastes, which influences and increases costs for treating sewage plants and polluted waters [17–20]. Biodiesel can be produced from

vegetable oils through a transesterification reaction that converted triglycerides into fatty acid methyl or ethyl esters using the low molecular weight alcohols (Scheme 1). This reaction can be carried out in different conditions and with different types of homogeneous and heterogeneous catalysts [3–5].

**Scheme 1.** Transesterification reaction.

Basic catalysis represents the most widely used industrial approach towards biodiesel production [21–23].

The advantages in using this type of catalysis concern reduced reaction times, moderate operating temperatures, the easy recovery of glycerol and the use of cost-effective catalysts. The main disadvantages relate to the saponification reaction, which entails lower yields and difficult washing of biodiesel, with excessive loss of product due to the formation of emulsions. Furthermore, an initial high content of free fatty acids (FFA) allows saponification reaction to undermine the final yield. The reaction was also conditioned by the water content of the sample, which can be reduced through an easy pretreatment such as the heating of the starting oil [24].

Otherwise, the advantage in using homogeneous acidic catalysis is related to the insensitivity to high levels of FFA, simultaneously promoting both the transesterification and the FFA esterification reactions, resulting into an improved biodiesel conversion yield [25,26].

The acidic catalysis results to be much more effective than the basic one when the vegetable oil contains more than 1% of free fatty acids. The transesterification process catalyzed by Brønsted acids, as sulfuric acid, allows very high yields in alkyl esters, however, the drawbacks are related to longer reaction times and higher temperatures [27,28].

Although the use of $H_2SO_4$ as a catalyst in biodiesel production was largely investigated, the transesterification of a highly pollutant starting material as waste cooking oils (WCO), catalyzed by sulfuric acid, was less explored.

Nye and coworkers reported that it is possible to produce biodiesel from exhausted frying oil using 0.1% of $H_2SO_4$ as a catalyst, methanol as a solvent, with a methanol/oil molar ratio of 3.6:1, at 25 °C, but the product was recovered with a yield of 72% after a reaction time of 40 h [29].

Canakci and Van Gerpen investigated the transesterification of food grade soybean oil, catalyzed by sulfuric acid, with a MeOH/oil molar ratio of 6:1, a reaction temperature of 60 °C, a catalyst amount of 3%, after a reaction time of 48 h. Nevertheless, the acid catalyst required a concentration of water lower than 0.5% to yield 88% of biodiesel [30].

Guana and Kusakabe developed the synthesis of biodiesel from waste oily sludge (WOS) with high free fatty acids (FFA) content. $H_2SO_4$ and $Fe_2(SO_4)_3$ were used as catalysts for the esterification of FFA, yielding biodiesel in 86% for both catalysts under suitable condition. The $H_2SO_4$ concentration was 3 wt% and MeOH/oil molar ratio was 10:1. The reaction rate was greatly enhanced by increasing the reaction temperature to 100 °C. However, this method required a preliminary extraction of oil from WOS containing water, using hexane [31].

In addition to homogeneous catalysis, the heterogeneous catalysis, according to the objectives of green chemistry, provides the facility of separation of the product and the catalyst, allowing the recovery and reuse of the catalyst [32]. The active catalytic material is often dispersed on the surface of a

support that can be catalytically inert or can contribute to the overall catalytic activity. Typical supports include different materials such as activated carbon, black carbon and carbon nanotubes, many types of porous inorganic solids such as silica, zeolites, titanium dioxide and mesoporous materials [33].

In this work the main objective concerned the optimization of the transesterification reaction with a focus on acidic catalysis, using different waste cooking oils as starting material. In this study we offer complementary approaches towards biodiesel production from real and unknown mixtures of exhausted frying oils. The advantage of the proposed procedure consists in the robustness of the method with respect to mixtures of very different exhausted oils.

## 2. Results and Discussion

### 2.1. Basic Catalysis

We firstly explored the transesterification reaction using basic catalysis, to evaluate the composition of the fatty acids in the starting mixture. The experiments were carried out on six samples, two of which consisted exclusively of waste olive and seed oils, while four consisted of unknown waste oil mixtures. The WCO blends were heated for five hours at a temperature of 65 °C to eliminate impurities and excess of water [24].

The heated samples were then subjected to the transesterification reaction with methanol using NaOH 1% *w/w* as catalyst, stirring the mixtures at a temperature of 50 °C for 1 h, with a molar ratio of MeOH/oil of 6:1. All the methyl esters were obtained with very high conversion and the separation of glycerol from the methyl esters mixture easily occurred. The results are shown in Table 1.

**Table 1.** Optimization study of the transesterification in homogeneous basic catalysis.

| Entry | Source | Crude weight [1] (g) | Conversion [2] (%) |
|:---:|:---:|:---:|:---:|
| 1 | WCO olive oil | 68.4 [a] | 94 |
| 2 | WCO seed oil | 65.4 [a] | >99 |
| 3 | WCO mix 1 | 73.0 [a] | 97 |
| 4 | WCO mix 2 | 62.0 [a] | 91 |
| 5 | WCO mix 3 | 4.4 [b] | >99 |
| 6 | WCO mix 4 | 73.0 [a] | >99 |

Conditions: NaOH 1% *w/w*, MeOH/oil ratio 6:1, 50 °C, 1h. [1] Weight of recovered biodiesel: [a] weight of the initial waste cooking oil (WCO) = 100 g; [b] weight of the initial WCO = 10 g. [2] Calculated by [1]H-NMR.

### 2.2. Acidic Catalysis

A screening was performed using concentrated $H_2SO_4$ (20% *w/w*) as a catalyst. Therefore, various reaction parameters were modified to test their influence on biodiesel yield. Table 2 showed the experiments performed using all the waste oils as the starting material.

At first the reaction was carried out on all the WCO mixtures, at a temperature of 65 °C, for 12 h, with MeOH/oil molar ratios of 6:1, reaching moderate biodiesel conversion (entries 1–6). Using WCO mix 2 as the model mixture, the temperature was raised to 85 °C and the reaction time was reduced to 6 h, while the MeOH/oil ratio was also modified. As a result, higher temperature enhanced the conversion of WCO mix 2 into biodiesel. Setting MeOH/oil ratio to 3:1 and 6:1 (entries 7–8) biodiesel was afforded with comparable conversion, which dropped when 12:1 MeOH/oil ratio was used (entry 9). By increasing the reaction time to 12 h at 85 °C, conversion has reached optimum levels up to 99% (entries 13 and 20). To investigate the scalability of the process, the reaction was scaled up using 100 g of the starting WCO mix 2 with MeOH/oil ratio of 3:1 and 6:1, and in both cases (entries 14 and 21) the conversion was found to be excellent (96% and 99% respectively). Furthermore, all the WCO mixtures were transformed into biodiesel by changing MeOH/oil ratios and temperature. The alcohol/oil molar ratio is one of the main factors that influence transesterification. An excess of the alcohol promotes the formation of the biodiesel, but the ideal alcohol/oil ratio must be established empirically, considering

each individual process [30]. As a result, biodiesel was obtained with the highest conversion from all the mixtures, heating the samples at 85 °C, with a reaction time of 12 h and 6:1 MeOH/oil ratio.

**Table 2.** Optimization study of the transesterification in homogeneous acidic catalysis.

| Entry | Source | Temperature (°C) | Time (h) | MeOH/Oil Ratio | Crude Weight [1] (g) | Conversion [2] (%) |
|---|---|---|---|---|---|---|
| 1 | WCO olive oil | 65 | 12 | 6:1 | 8.3 [a] | 66 |
| 2 | WCO seed oil | 65 | 12 | 6:1 | 6.3 [a] | 63 |
| 3 | WCO mix 1 | 65 | 12 | 6:1 | 7.5 [a] | 45 |
| 4 | WCO mix 2 | 65 | 12 | 6:1 | 6.2 [a] | 48 |
| 5 | WCO mix 3 | 65 | 12 | 6:1 | 7.2 [a] | 34 |
| 6 | WCO mix 4 | 65 | 12 | 6:1 | 7.1 [a] | 40 |
| 7 | WCO mix 2 | 85 | 6 | 3:1 | 7.4 [a] | 75 |
| 8 | WCO mix 2 | 85 | 6 | 6:1 | 4.2 [a] | 72 |
| 9 | WCO mix 2 | 85 | 6 | 12:1 | 9.6 [a] | 42 |
| 10 | WCO olive oil | 85 | 12 | 3:1 | 5.4 [a] | 79 |
| 11 | WCO seed oil | 85 | 12 | 3:1 | 6.3 [a] | 74 |
| 12 | WCO mix 1 | 85 | 12 | 3:1 | 9.3 [a] | 68 |
| 13 | WCO mix 2 | 85 | 12 | 3:1 | 7.2 [a] | 98 |
| 14 | WCO mix 2 | 85 | 12 | 3:1 | 72.0 [b] | 96 |
| 15 | WCO mix 3 | 85 | 12 | 3:1 | 7.5 [a] | 69 |
| 16 | WCO mix 4 | 85 | 12 | 3:1 | 7.9 [a] | 85 |
| 17 | WCO olive oil | 85 | 12 | 6:1 | 6.8 [a] | 92 |
| 18 | WCO seed oil | 85 | 12 | 6:1 | 8.5 [a] | 85 |
| 19 | WCO mix 1 | 85 | 12 | 6:1 | 8.4 [a] | 91 |
| 20 | WCO mix 2 | 85 | 12 | 6:1 | 6.6 [a] | 99 |
| 21 | WCO mix 2 | 85 | 12 | 6:1 | 88.0 [b] | 99 |
| 22 | WCO mix 3 | 85 | 12 | 6:1 | 9.9 [a] | 86 |
| 23 | WCO mix 4 | 85 | 12 | 6:1 | 8.9 [a] | 91 |
| 24 | WCO olive oil | 85 | 12 | 12:1 | 8.0 [a] | 61 |
| 25 | WCO seed oil | 85 | 12 | 12:1 | 7.5 [a] | 63 |
| 26 | WCO mix 1 | 85 | 12 | 12:1 | 8.1 [a] | 35 |
| 27 | WCO mix 2 | 85 | 12 | 12:1 | 7.6 [a] | 70 |
| 28 | WCO mix 3 | 85 | 12 | 12:1 | 7.4 [a] | 37 |
| 29 | WCO mix 4 | 85 | 12 | 12:1 | 7.2 [a] | 60 |

[1] Weight of recovered biodiesel: [a] weight of the initial WCO = 10 g; [b] weight of the initial WCO = 100 g. [2] Conversion calculated by $^1$H-NMR.

From the experimental point of view, the acid catalyzed reactions presented several problems related to the recovery of glycerol and product washing, compared to the basic catalyzed ones. The addition of concentrated $H_2SO_4$ to the reaction mixture caused a change of color, from yellow to black. It was hypothesized that this variation could be attributable to the formation of polymeric products [34]. In each transesterification reaction catalyzed by $H_2SO_4$, the formation of the dark amorphous residue was observed after some hours. This residue was insoluble in the reaction mixture and in water, whereas it was found to be soluble in acetone. Biodiesel washing represented another problem related to the use of this acid catalyst. The presence of the dark residue, in fact, makes it difficult to observe the phase separation during the work up procedure. Furthermore, an evaluation of acidic catalysis was carried out, reporting the role of silica as an element introduced directly into the reaction mixture [34]. In recent years, the development of new catalytic strategies in the production of biodiesel has witnessed an increase in the use of silica as a support for charging acid sulfonic catalysts, which have shown promising catalytic activity [35]. The use of silica was evaluated to cope with a further problem arising during the catalyzed acid transesterification reactions with $H_2SO_4$. These reactions were characterized by the formation of black residues, water-insoluble and hard to remove. The tests were conducted using WCO mix 2, as reference, with the addition of 50% *w/w* of silica (Table 3).

**Table 3.** Optimization study of the transesterification of WCO mix 2 in the presence of silica.

| Entry | Catalyst | Time (h) | Crude weight [1] (g) | Conversion [2] (%) |
|---|---|---|---|---|
| 1 | $SiO_2$ | 6 | - | - |
| 2 | $SiO_2$/Amberlyst 15 (3%) | 6 | - | - |
| 3 | $SiO_2$/HCl 4M in dioxane | 6 | 5.3 [a] | 16 |
| 4 | $SiO_2$/$H_2SO_4$ (20%) | 6 | 8.7 [a] | 23 |
| 5 | $SiO_2$/$H_2SO_4$ (20%) | 12 | 8.5 [b] | 92 |
| 6 | $SiO_2$/$H_2SO_4$ (20%) | 12 | 10.5 + 5.3 [b,c] | 85 |

[1] Weight of recovered biodiesel: [a] weight of the initial WCO = 10 g; [b] weight of the initial WCO = 20 g; [c] after extraction with $CH_2Cl_2$. [2] Calculated by $^1$H-NMR.

At first the reactions carried out with silica and a cation exchange resin such as Amberlyst 15 at 3% *w/w* (after activation with HCl 6N) were tested (entries 1 and 2), without any biodiesel conversion. Afterwards, using silica with HCl 4M in dioxane and $H_2SO_4$ 20% *w/w* (entries 3 and 4) after 6 h, low conversions were observed while the best result was obtained stirring the reaction for 12 h (entry 5). However, part of the obtained biodiesel was still retained by silica, therefore supplementary extraction of the retained material on silica with $CH_2Cl_2$ allowed the recovery of additional 5.3 g of biodiesel (entry 6). These results were accompanied by the good ability of silica to retain the residues formed during the reaction.

*2.3. Characterization*

2.3.1. Acidity of WCO

The acidity of the waste oils was a fundamental parameter, as it influenced the yield of the transesterification reaction. By titration of WCO samples, it was possible to verify that free fatty acids content was lower than 3%, except of WCO mix 1 that showed an acidity of 3.16%. The analyzed samples were heated to 65 °C for 5 h and the acidity percentage was measured before and after heating. The percentage acidity values calculated for different oil mixtures are shown in Table 4.

**Table 4.** Percentage acidity values.

| Entry | Source | Acidity % [1] | Acidity % [2] |
|---|---|---|---|
| 1 | WCO olive oil | 1.79 | 1.47 |
| 2 | WCO seed oil | 1.73 | 1.61 |
| 3 | WCO mix 1 | 3.16 | 2.80 |
| 4 | WCO mix 2 | 0.68 | 0.84 |
| 5 | WCO mix 2 [3] | 0.68 | 0.57 |
| 6 | WCO mix 3 | 2.05 | 2.22 |
| 7 | WCO mix 3 [3] | 2.05 | 2.02 |
| 8 | WCO mix 4 | 0.34 | 0.31 |

[1] Before heating. [2] After heating. [3] Dried with anhydrous $Na_2SO_4$ before heating.

The acidity decreased after heating, except in two cases (entries 4 and 6) where the acidity was higher with respect to the starting sample. To justify these results, it was hypothesized that the presence of a high amount of water in the starting WCO may increase the concentration of acidic species after heating. To eliminate the excess of water, WCO mix 2 and WCO mix 3 were preliminarily dried with anhydrous $Na_2SO_4$. In this case, repeating the titration after the thermal treatment (entries 5 and 7), the acidity percentage decreased. Consequently, it was possible to state that water could have a significant role in lowering the transesterification reaction yield, due to the thermal promoted hydrolysis of triglycerides of WCO mixtures.

### 2.3.2. NMR Characterization

The starting material was analyzed by NMR spectroscopy and the analysis of [1]H-NMR spectra is reported in Figure 1. The signal at 5.35 ppm corresponds to the olefinic hydrogen of unsaturated fatty acids. The signal at 5.20 ppm corresponds to glycerol protons while glycerol methylene protons showed a signal at 4.19 ppm. Allylic protons showed a signal at 1.88–2.02 ppm and 2.63–2.77 ppm, signals at 2.15–2.31 and 1.47–1.62 ppm corresponded to methylene protons in α e β in the carboxy group, while the saturated groups exhibited a signal at 1.14–1.26 ppm. Signals at 0.75–0.88 and 0.95 ppm corresponded to the methyl protons of the aliphatic chains of fatty acids. A peak at a chemical shift value of 2.78 ppm, corresponding to the signal of linoleic acid and particularly abundant in seed oil, was also detected. All the mixtures were analyzed and a comparison of the composition of starting oil blends was performed, as shown in Figure 2. The transesterification reaction was verified from the analysis of the [1]H-NMR spectra of the corresponding biodiesel samples and the conversion was calculated according to the literature [36]. In fact, it was possible to detect the presence of a peak at a chemical shift value of 3.67 ppm ($A_1$), identifying the presence of the methoxy group of fatty acid esters while the signals of the methylene groups at 5.20 and 4.19 ppm ($A_2$) disappeared (Figure 3).

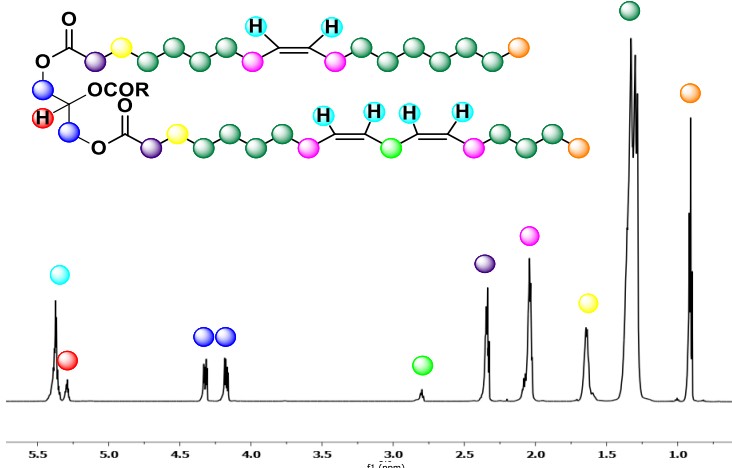

**Figure 1.** [1]H-NMR spectra of a general WCO mixture.

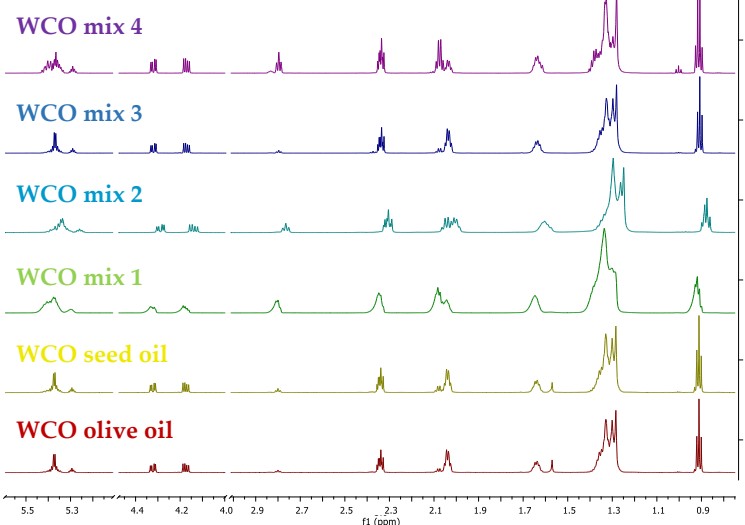

**Figure 2.** [1]H-NMR spectra of WCO mixtures.

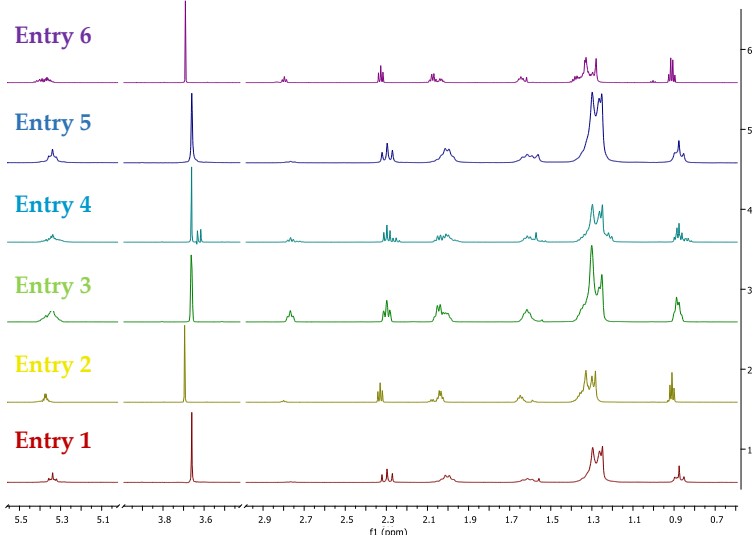

**Figure 3.** $^1$H-NMR spectra of fatty acid methyl esters (FAMEs) obtained with basic catalysis (Table 1).

### 2.3.3. FT-IR Analysis

FT-IR spectroscopy represented a rapid and precise method for the recognition of functional groups of fatty acids deriving from WCO and biodiesel [37].

The main difference between the two FT-IR spectra was related to the transformation of the exhausted oil ester groups into the methyl esters of the produced biodiesel (Supporting Information). The exhausted vegetable oils showed an absorption that appears at 722 cm$^{-1}$, representative of the rocking band of the –CH$_2$ and the other at 1746 cm$^{-1}$ typical of the stretching of the C=O of the ester. Biodiesel absorptions appeared at 1464 and 1437 cm$^{-1}$, representing the methyl ester groups, and at 1196 cm$^{-1}$, corresponding to –(C–O)– of the ester. The peak at 1163 cm$^{-1}$ of the oil sample compared to the 1195 cm$^{-1}$ peak of the biodiesel sample demonstrated the successful conversion.

### 2.3.4. Gas Chromatographic Analysis

Gas chromatography allowed the estimation of peaks obtained by the analysis of the methyl esters derived from the fatty acids after transesterification. Consequently, it was also possible to trace the composition of the fatty acids present in the triglycerides of the starting material [38].

In fact, the area underneath these peaks provided information about the related content of a specific methyl ester in the mixture (Table 5).

**Table 5.** Composition of FAMEs.

| Entry | Source | Palmitoleate | Palmitate | Linoleate | Oleate | Stearate |
|-------|--------------|------------|---------|---------|--------|--------|
| 1 | WCO olive oil | 1.441 | 12.284 | 8.701 | 75.255 | 2.319 |
| 2 | WCO seed oil | 1.324 | 12.015 | 13.430 | 70.655 | 2.574 |
| 3 | WCO mix 1 | - | 6.908 | 53.366 | 35.618 | 4.108 |
| 4 | WCO mix 2 | - | 7.484 | 37.064 | 52.017 | 3.434 |
| 5 | WCO mix 3 | 2.400 | 13.046 | 10.163 | 71.785 | 2.515 |
| 6 | WCO mix 4 | - | 9.027 | 55.759 | 29.864 | 5.350 |

## 3. Materials and Methods

### 3.1. Chemicals and Material

All the reactions were performed on a series of samples of waste oils for domestic use: waste olive oil, waste seed oil, mixture of different waste oils (WCO mix 1, WCO mix 2 and WCO mix 3), and

mixture of different waste oils used at a constant temperature of 180 °C (WCO mix 4). All the other chemicals were commercially available and used without further purification.

### 3.2. General Reaction Procedures

The molecular weight of the WCO mixtures was estimated considering the composition of the lateral chains $R_1$, $R_2$ and $R_3$ deriving from oleic, linoleic and palmitic acid respectively, as confirmed by Gas Chromatographic analysis.

### 3.2.1. Homogeneous Basic Catalysis

In a 250 mL round bottom flask, 100 g of WCO (117 mmol) were refluxed at 65 °C for five hours. Of MeOH (624 mmol) 20 g was mixed with 1 g of NaOH (25 mmol), the solution was added to the oil and the reaction was refluxed at 50 °C for one hour. The organic phase was washed with $NH_4Cl$ and distilled water, dried with $Na_2SO_4$, filtered and finally concentrated under vacuum.

### 3.2.2. Homogeneous Acidic Catalysis

In a 50 mL round bottom flask, 10 g of WCO (11.7 mmol) were mixed with 2 g of MeOH (62.4 mmol). Finally, 1.08 mL of concentrated $H_2SO_4$ (20 mmol) was added. The reaction was refluxed at a temperature of 85 °C for of twelve hours. The organic phase was extracted with water and diethyl ether, dried with $Na_2SO_4$, filtered and finally concentrated under vacuum.

### 3.2.3. Acidic Catalysis Supported by Silica

In a 100 mL round bottom flask, 20 g of WCO mix 2 (23.3 mmol) were mixed with 4 g of MeOH (125 mmol) and 10 g of gravitational $SiO_2$ (166 mmol). Finally, 2.17 mL (41 mmol) of concentrated $H_2SO_4$ were added. The reaction was refluxed at a temperature of 85 °C for 12 h. The product was filtered under vacuum by using a Gooch filter. The organic phase was washed with $CH_2Cl_2$, filtered and finally concentrated under vacuum.

### 3.3. Acidity Percentage

The mixture of oil (10 g), ethanol 95° (100 g), diethyl ether (300 g) and phenolphthalein (2 mL) was titrated with KOH (0.1 M), until a color change from colorless to pale pink. The mL of KOH needed to reach the color change point is included in the following formula to calculate percentage acidity:

$$\%A = \frac{M \times V \times MW_{OL}}{P \times 10} \times 100 \tag{1}$$

V = volume of KOH at the turning point;
M = KOH molarity;
$MW_{OL}$ = molecular weight of oleic acid;
P = weight in g of oil used for titration.

### 3.4. Analysis Methods

The [1]H-NMR spectra were recorded with a VARIAN INOVA 500 MHz or Bruker 700 MHz spectrometer; all chemical shift values are reported in ppm (δ); $CDCl_3$ was used as solvent.

The conversion was calculated from the ratio between the selected areas of $A_1$ (methoxy group of FAME, 3.67 ppm, singlet) and $A_2$ (methylene group of WCO, 5.20 or 4.19 ppm, dd) according to the following equation [36]:

$$\text{Conversion (\%)} = 100 \times (2A_1/3A_2). \tag{2}$$

FT-IR spectra were performed with a PERKIN ELMER 283 spectrophotometer. The gas chromatograms were obtained with a HP6890 plus gas chromatograph equipped with HP1 column.

## 4. Conclusions

Hence, it could be concluded that WCO, which is economic and easily available, is a very useful raw material for biodiesel production. The optimization of a transesterification reaction of six different waste oil mixtures was carried out. The results obtained have shown that the basic catalysis allowed us to evaluate the composition of the fatty acids in the starting mixture. The conditions of acidic catalysis have therefore been widely explored. Under optimized conditions all the mixtures show very high conversions. However, the recovery of glycerol and crude washing were difficult, due to the formation of insoluble products probably deriving from oxidation processes. To overcome this drawback, the use of silica was tested. As a result, in the presence of silica as supporting material, biodiesel was prepared with high conversion, whereas the insoluble material was retained on the silica surface. Furthermore, the role of silica facilitated the purification process because the formation of polymeric materials can contaminate the purity of the formed biodiesel.

**Supplementary Materials:** The following are available online at http://www.mdpi.com/2073-4344/9/12/1050/s1, Figure S1: $^1$H-NMR spectra of FAMEs from WCO olive oil (Table 2), Figure S2: $^1$H-NMR spectra of FAMEs from WCO seed oil (Table 2), Figure S3: $^1$H-NMR spectra of FAMEs from WCO mix 2 (Table 2), Figure S4: $^1$H-NMR spectra of FAMEs from WCO mix 1 (Table 2), Figure S5: $^1$H-NMR spectra of FAMEs from WCO mix 3 (Table 2), Figure S6: $^1$H-NMR spectra of FAMEs from WCO mix 4 (Table 2), Figure S7: $^1$H-NMR spectra of FAMEs from WCO mix 2 (Table 3), Figure S8: $^1$H-NMR spectrum of standard biodiesel, Figure S9: FT-IR spectrum of WCO olive oil, Figure S10: FT-IR spectrum of WCO seed oil, Figure S11: FT-IR spectrum of WCO mix 1, Figure S12: FT-IR spectrum of WCO mix 2, Figure S13: FT-IR spectrum of WCO mix 3, Figure S14: FT-IR spectrum of WCO mix 4, Figure S15: FT-IR spectrum of Entry 1 (Table 1), Figure S16: FT-IR spectrum of Entry 2 (Table 1), Figure S17: FT-IR spectrum of Entry 3 (Table 1), Figure S18: FT-IR spectrum of Entry 4 (Table 1), Figure S19: FT-IR spectrum of Entry 5 (Table 1), Figure S20: FT-IR spectrum of Entry 6 (Table 1), Figure S21: Chromatogram of Entry 1 (Table 5), Figure S22: Chromatogram of Entry 2 (Table 5), Figure S23: Chromatogram of Entry 3 (Table 5), Figure S24: Chromatogram of Entry 4 (Table 5), Figure S25: Chromatogram of Entry 5 (Table 5), Figure S26: Chromatogram of Entry 6 (Table 5), Figure S27: Chromatogram of Standard Biodiesel sample, Table S1: GC parameters.

**Author Contributions:** C.C. conceived, designed, performed the experiments and wrote the paper; M.A. analyzed the data and wrote the paper; R.L. and L.D. contributed reagents/materials/analysis tools.

**Funding:** This research received no external funding.

**Acknowledgments:** This work was supported by the *Intervento cofinanziato dal Fondo di Sviluppo e Coesione 2007–2013–APQ Ricerca Regione Puglia "Programma regionale a sostegno della specializzazione intelligente e della sostenibilità sociale ed ambientale-FutureInResearch"*. Saverio Cellamare and Cosimo Damiano Altomare are acknowledged for their technical and logistics support. We are also grateful to Rebecca Ostuni for the precious contribution.

**Conflicts of Interest:** The authors declare no conflict of interest.

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
