# Peer review of "Benchmarking Acidic and Basic Catalysis for a Robust Production of Biofuel from Waste Cooking Oil"

_catalysts, doi:10.3390/catal9121050_

Round 1

Reviewer 1 Report

The manuscript lacks novelty. It can be considered an incremental work where different type of oil waste materials are converted into biodiesel under already reported conditions

Author Response

Aware on the point of view of the referee, we changed the manuscript and added details to justify the advantages of the proposed methodology.

Reviewer 2 Report

Comments to the paper: “Benchmarking Acid and Basic Catalysis for a Robust Production of Biofuel from Waste Cooking Oil” by Carlucci et al.

General comment:

The paper deals with the investigations of catalytic pathways in the production of FAMEs, by testing different cooking waste oil mixtures and exploring homogeneous and heterogenous acid catalysis. In particular, the effects of catalyst type, temperature, reaction time and MeOH/oil molar ratios were studied.

The manuscript is suitable to be published in this journal; however some points should be addressed before publication. Some minor language mistakes are present that should anyway be corrected.

1. Introduction

Please improve Introduction by including recent reviews or research article focused on FAMEs production. I would like to suggest to include the following papers into the literature overview:

Leone, G.P., Balducchi, R., Mehariya, S., Martino, M., Larocca, V., Sanzo, G.D., Iovine, A., Casella, P., Marino, T., Karatza, D., Chianese, S., Musmarra, D., Molino, A., Selective extraction of ω-3 fatty acids from nannochloropsis sp. using supercritical CO2 Extraction (2019) Molecules, 24 (13), art. no. 2406. Machmudah, S., Wahyudionob, Kanda, H., Goto, M., 2018. Supercritical fluids extraction of valuable compounds from algae: Future perspectives and challenges, Engineering Journal, 22(5), 13-30. Di Sanzo, G., Mehariya, S., Martino, M., Larocca, V., Casella, P., Chianese, S., Musmarra, D., Balducchi, R., Molino, A. Supercritical carbon dioxide extraction of astaxanthin, lutein, and fatty acids from haematococcus pluvialis microalgae (2018) Marine Drugs, 16 (9), art. no. 334. Catchpole, O., Moreno, T., Montañes, F., Tallon, S., 2018. Perspectives on processing of high value lipids using supercritical fluids, Journal of Supercritical Fluids, 134, 260-268. Schablitzky, H.W., Lichtscheidl, J., Hutter, K., Hafner, C., Rauch, R., Hofbauer, H. Hydroprocessing of Fischer-Tropsch biowaxes to second-generation biofuels (2011) Biomass Conversion and Biorefinery, 1 (1), pp. 29-37. Materials and Methods

Please, clarify if waste cooking oil/catalyst molar ratio was varied; if so, please specify the variation range.

According to your procedure it seems that different ratios between waste cooking oil and catalyst were used depending on the type of catalyst and the type of reaction, instead of use the same variation range, and the same weight, could you please argument this procedure?

Please, specify if investigations were carried out in duplicate/triplicate etc.

Results and Discussion

Please, improve comparison between experimental findings and literature data.

I would like to suggest to compare the characteristics/standard of the produced biodiesel with the ones of commercial diesel.

Author Response

The paper deals with the investigations of catalytic pathways in the production of FAMEs, by testing different cooking waste oil mixtures and exploring homogeneous and heterogenous acid catalysis. In particular, the effects of catalyst type, temperature, reaction time and MeOH/oil molar ratios were studied. The manuscript is suitable to be published in this journal; however some points should be addressed before publication. Some minor language mistakes are present that should anyway be corrected.

Response: We thank the Reviewer for his comments and criticisms. In agreement with the suggestions we have modified the manuscript accordingly and we have replied point counter point to the remarks, as reported below.

Point 1: Introduction

Please improve Introduction by including recent reviews or research article focused on FAMEs production. I would like to suggest to include the following papers into the literature overview:

Leone, G.P., Balducchi, R., Mehariya, S., Martino, M., Larocca, V., Sanzo, G.D., Iovine, A., Casella, P., Marino, T., Karatza, D., Chianese, S., Musmarra, D., Molino, A., Selective extraction of ω-3 fatty acids from nannochloropsis sp. using supercritical CO2 Extraction (2019) Molecules, 24 (13), art. no. 2406. Machmudah, S., Wahyudionob, Kanda, H., Goto, M., 2018. Supercritical fluids extraction of valuable compounds from algae: Future perspectives and challenges, Engineering Journal, 22(5), 13-30. Di Sanzo, G., Mehariya, S., Martino, M., Larocca, V., Casella, P., Chianese, S., Musmarra, D., Balducchi, R., Molino, A. Supercritical carbon dioxide extraction of astaxanthin, lutein, and fatty acids from haematococcus pluvialis microalgae (2018) Marine Drugs, 16 (9), art. no. 334. Catchpole, O., Moreno, T., Montañes, F., Tallon, S., 2018. Perspectives on processing of high value lipids using supercritical fluids, Journal of Supercritical Fluids, 134, 260-268. Schablitzky, H.W., Lichtscheidl, J., Hutter, K., Hafner, C., Rauch, R., Hofbauer, H. Hydroprocessing of Fischer-Tropsch biowaxes to second-generation biofuels (2011) Biomass Conversion and Biorefinery, 1 (1), pp. 29-37.

Response 1: The suggested references were added, accordingly.

Point 2: Materials and Methods

Please, clarify if waste cooking oil/catalyst molar ratio was varied; if so, please specify the variation range. According to your procedure it seems that different ratios between waste cooking oil and catalyst were used depending on the type of catalyst and the type of reaction, instead of use the same variation range, and the same weight, could you please argument this procedure? Please, specify if investigations were carried out in duplicate/triplicate etc.

Response 2:  the molar ratio between waste cooking oil and the acid or basic catalysts remained constant in both the procedures. The basic catalysis was explored to identify the composition of the FAMEs because of the high conversion obtained in all the tests and the easy recovery of the product, but no comparison between the two catalytic processes was investigated. The investigations were duplicated in the optimized conditions and the reproducibility of the results was verified.

Point 3: Results and Discussion

Please, improve comparison between experimental findings and literature data. I would like to suggest to compare the characteristics/standard of the produced biodiesel with the ones of commercial diesel.

Response 3: we compared the NMR spectrum and the chromatogram of a standard biodiesel sample with our samples and the results obtained were comparable. The corresponding spectra were added to the supporting information.

Reviewer 3 Report

Dear Authors,

Please follow the next observations:

a nomenclature will be appreciated; please add some quantitative values and authoritative conclusions in the abstract;
please present the downsides and advantages of the proposes method in comparison with others; add axes titles to the figures; the conclusion section is to trivial.Best regards!

Author Response

Please follow the next observations:

Response: We thank the Reviewer for his comments and criticisms. In agreement with the observations we have modified the manuscript accordingly and we have replied point counter point to the remarks, as reported below.

Point 1: a nomenclature will be appreciated;

Response 1: the composition of WCO was estimated but it is not possible to attribute the exact nomenclature to the mixtures.

Point 2: please add some quantitative values and authoritative conclusions in the abstract.

Response 2:  the abstract was improved, accordingly.

Point 3: please present the downsides and advantages of the proposes method in comparison with others.

Response 3:  the downsides and advantages were added in the discussion.

Point 4: add axes titles to the figures.

Response 4:  The figures were modified accordingly.

Point 5: the conclusion section is to trivial.

Response 5:  The conclusions have been improved, accordingly.

Reviewer 4 Report

After I read the article carefully, I think that the research is carried out correctly and rigorously, and the authors provide data supported by the graphs, chromatograms ... however, I would like to discuss with the authors some aspects of this piece of research.

First, table 1, the numbering presented in the table footer does not correspond to the numbering presented in the table. (conversion is 1, and 1 is weight of recovered biodiesel)

I would also like to ask the authors about the novelty and significance of this research, and what distinguishes it from other similar research.

The authors speak of heterogeneous catalysis, but it is not correct, because the catalyst used is H2SO4 and the role of SIO2 is to retain the black waste that is produced. Or the authors have tried to reuse that SiO2 without the addition of H2SO4 again.

Referring to table 3, entry 5 and 6, they have the same conditions, but at entry 5 it presents greater conversion and less crude weight before washing with Ch2Cl2. Why?

Table 2 should be explained more deeply for a better understanding. Influence of temperature and ratio on different WCO, and what may be the reason for these influences.

When performing the optimization tests of the different ratios, is the reaction volume always used the same? In the materials and methods section it is not specified.

Author Response

After I read the article carefully, I think that the research is carried out correctly and rigorously, and the authors provide data supported by the graphs, chromatograms ... however, I would like to discuss with the authors some aspects of this piece of research.

Response: We thank the Reviewer for his comments and criticisms. In agreement with the suggestions we have modified the manuscript accordingly and we have replied point counter point to the remarks, as reported below.

Point 1: First, table 1, the numbering presented in the table footer does not correspond to the numbering presented in the table. (conversion is 1, and 1 is weight of recovered biodiesel)

Response 1: the numbering has been corrected accordingly.

Point 2: I would also like to ask the authors about the novelty and significance of this research, and what distinguishes it from other similar research.

Response 2:  the novelty and significance of the research were added in the discussion, accordingly.

Point 3: The authors speak of heterogeneous catalysis, but it is not correct, because the catalyst used is H2SO4 and the role of SiO2 is to retain the black waste that is produced. Or the authors have tried to reuse that SiO2 without the addition of H2SO4 again.

Response 3:  as far as heterogeneous catalysis is concerned, we performed only one test with silica in the absence of a catalyst that produced no results (Table 3, Entry 1). Therefore, we have modified the title of the paragraph in accordance with the referee's suggestion.

Point 4: Referring to table 3, entry 5 and 6, they have the same conditions, but at entry 5 it presents greater conversion and less crude weight before washing with CH2Cl2. Why?

Response 4:  it was better clarified that it was not a washing procedure but an additional extraction of the retained material on the silica surface. We suppose that silica can be able to retain the starting WCO mixture with high affinity. Therefore, after the extraction, all the starting material was recovered thus lowering the estimated conversion.

Point 5: Table 2 should be explained more deeply for a better understanding. Influence of temperature and ratio on different WCO, and what may be the reason for these influences.

Response 5:  the influence of temperature and ratio was described, and the corresponding reference was added.

Point 6: When performing the optimization tests of the different ratios, is the reaction volume always used the same? In the materials and methods section it is not specified.

Response 6:  the reaction volume is always the same, keeping constant the amount of the starting material. When the reaction was carried out on 10 g of starting WCO a round bottom flask of 50 mL, while a round bottom flask of 250 mL was used for 100 g scaled reactions. The sentence was added in the experimental section accordingly.

Round 2

Reviewer 1 Report

The article presents a rigorous and well performance study for the synthesis of the biodiesel using waste cooking oils, the research also presents low degree of novelty although it may be of interest for specialist readers working in this particular field. Some minor changes should be introduced to gain in clarity.

The conversion are calculated by NMR and the NMR characterization a given in the section 2.3.2, but some things are missing. In any of the NMR figures, the solvent is not provided but in the experimental part, line 259 three different solvents are mention. Please specify the solvent use for figures 1, 2 and 3 and all NMRs of the SI. Please explain how the conversion is calculated by NMR what peaks are used? With or without and internal standard? Although a reference is provided (ref27), a more detailed explanation is required

Author Response

We thank the Reviewer for his comments and criticisms. In agreement with the observation the only solvent used was CDCl3 and the detail was corrected in the experimental part accordingly. The conversion was calculated without internal standard, directly obtained from the ratio between the selected areas of A1 (methoxy group of FAME) and A2 (methylene group of WCO) according to the corresponding equation, added in the section 3.4.

Reviewer 2 Report

The authors revised the manuscript according to the comments/changes suggested, however I would like to suggest to check the reference list (in particular new references) and reference number into the manuscript.

Author Response

We thank the Reviewer for his suggestion. We have checked the references and the corresponding numbers accordingly.

Reviewer 4 Report

The authors have responded correctly to my suggestions.

Author Response

We thank the Reviewer for his contribution.